# SnapFusion: Text-to-Image Diffusion Model on Mobile Devices within Two Seconds

**Yanyu Li**[1,2,†]    **Huan Wang**[1,2,†]    **Qing Jin**[1,†]    **Ju Hu**[1]    **Pavlo Chemerys**[1]
**Yun Fu**[2]    **Yanzhi Wang**[2]    **Sergey Tulyakov**[1]    **Jian Ren**[1,†]
[1]Snap Inc.    [2]Northeastern University
Project Page: https://snap-research.github.io/SnapFusion

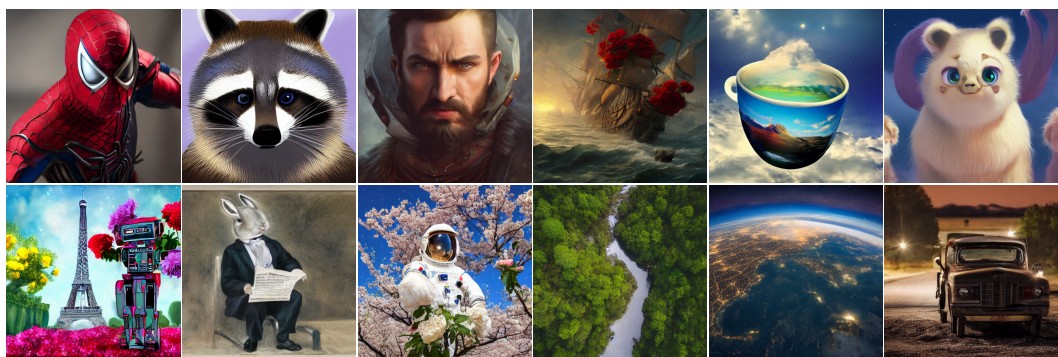

Figure 1: Example generated images by using our efficient text-to-image diffusion model.

## Abstract

Text-to-image diffusion models can create stunning images from natural language descriptions that rival the work of professional artists and photographers. However, these models are large, with complex network architectures and tens of denoising iterations, making them computationally expensive and slow to run. As a result, high-end GPUs and cloud-based inference are required to run diffusion models at scale. This is costly and has privacy implications, especially when user data is sent to a third party. To overcome these challenges, we present a generic approach that, for the first time, unlocks running text-to-image diffusion models on mobile devices in **less than 2 seconds**. We achieve so by introducing efficient network architecture and improving step distillation. Specifically, we propose an efficient UNet by identifying the redundancy of the original model and reducing the computation of the image decoder via data distillation. Further, we enhance the step distillation by exploring training strategies and introducing regularization from classifier-free guidance. Our extensive experiments on MS-COCO show that our model with 8 denoising steps achieves better FID and CLIP scores than Stable Diffusion v1.5 with 50 steps. Our work democratizes content creation by bringing powerful text-to-image diffusion models to the hands of users.

## 1  Introduction

Diffusion-based text-to-image models [1, 2, 3, 4] show remarkable progress in synthesizing pho-torealistic content using text prompts. They profoundly impact the content creation [5, 6], image

---

[†]Equal contribution.

37th Conference on Neural Information Processing Systems (NeurIPS 2023).

editing and in-painting [7, 8, 9, 10, 11], super-resolution [12], video synthesis [13, 14], and 3D assets generation [15, 16, 17], to name a few. This impact comes at the cost of the substantial increase in the computation requirements to run such models [18, 19, 20, 21]. As a result, to satisfy the necessary latency constraints large scale, often cloud-based inference platforms with high-end GPU are required. This incurs high costs and brings potential privacy concerns, motivated by the sheer fact of sending private images, videos, and prompts to a third-party service.

Not surprisingly, there are emerging efforts to speed up the inference of text-to-image diffusion models on mobile devices. Recent works use quantization [22, 23] or GPU-aware optimization to reduce the run time, *i.e.*, accelerating the diffusion pipeline to 11.5s on Samsung Galaxy S23 Ultra [24]. While these methods effectively achieve a certain speed-up on mobile platforms, the obtained latency does not allow for a seamless user experience. Besides, *none of the existing studies* systematically examine the generation quality of on-device models through quantitative analysis.

In this work, we present the first text-to-image diffusion model that generates an image on mobile devices in less than 2 seconds. To achieve this, we mainly focus on improving the slow inference speed of the UNet and reducing the number of necessary denoising steps. **First**, the *architecture of UNet*, which is the major bottleneck for the conditional diffusion model (as we show in Tab. 1), is rarely optimized in the literature. Existing works primarily focus on post-training optimizations [25, 26]. Conventional compression techniques, *e.g.*, model pruning [27, 28] and architecture search [29, 30], reduce the performance of pre-trained diffusion models [31], which is difficult to recover without heavy fine-tuning. Consequently, the architecture redundancies are not fully exploited, resulting in a limited acceleration ratio. **Second**, the flexibility of the *denoising diffusion process* is not well explored for the on-device model. Directly reducing the number of denoising steps impacts the generative performance, while progressively distilling the steps can mitigate the impacts [32, 33]. However, the learning objectives for step distillation and the strategy for training the on-device model have yet to be thoroughly studied, especially for models trained using large-scale datasets.

This work proposes a series of contributions to address the aforementioned challenges:

- We provide an in-depth analysis of the denoising UNet and identify the architecture redundancies.
- We propose a novel evolving training framework to obtain an efficient UNet that performs better than the original Stable Diffusion v1.5[1] while being significantly faster. We also introduce a data distillation pipeline to compress and accelerate the image decoder.
- We improve the learning objective during step distillation by proposing additional regularization, including losses from the **v**-prediction and classifier-free guidance [34].
- Finally, we explore the training strategies for step distillation, especially the best teacher-student paradigm for training the on-device model.

Through the improved **S**tep distillation and **n**etwork **a**rchitecture develo**p**ment for the dif**Fusion** model, our introduced model, **SnapFusion**, generates a $512 \times 512$ image from the text on mobile devices in *less than 2 seconds*, while with image quality similar to Stable Diffusion v1.5 [4] (see example images from our approach in Fig. 1).

## 2 Model Analysis of Stable Diffusion

### 2.1 Prerequisites of Stable Diffusion

**Diffusion Models** gradually convert the sample $\mathbf{x}$ from a real data distribution $p_{\text{data}}(\mathbf{x})$ into a noisy version, *i.e.*, the diffusion process, and learn to reverse this process by denoising the noisy data step by step [35]. Therefore, the model transforms a simple distribution, *e.g.*, random Gaussian noise, to the desired more complicated distribution, *e.g.*, real images. Specifically, given a (noise-prediction) diffusion model $\hat{\epsilon}_{\boldsymbol{\theta}}(\cdot)$ parameterized by $\boldsymbol{\theta}$, which is typically structured as a UNet [36, 1], the training can be formulated as the following noise prediction problem [35, 1, 2]:

$$\min_{\boldsymbol{\theta}} \ \mathbb{E}_{t \sim U[0,1], \mathbf{x} \sim p_{\text{data}}(\mathbf{x}), \boldsymbol{\epsilon} \sim \mathcal{N}(\mathbf{0}, \mathbf{I})} \ ||\hat{\epsilon}_{\boldsymbol{\theta}}(t, \mathbf{z}_t) - \boldsymbol{\epsilon}||_2^2, \tag{1}$$

where $t$ refers to the time step; $\boldsymbol{\epsilon}$ is the ground-truth noise; $\mathbf{z}_t = \alpha_t \mathbf{x} + \sigma_t \boldsymbol{\epsilon}$ is the noisy data; $\alpha_t$ and $\sigma_t$ are the strengths of signal and noise, respectively, decided by a noise scheduler. A trained

---

[1] https://github.com/runwayml/stable-diffusion

diffusion model can generate samples from noise with various samplers. In our experiments, we use DDIM [37] to sample with the following *iterative* denoising process from $t$ to a previous time step $t'$,

$$\mathbf{z}_{t'} = \alpha_{t'} \frac{\mathbf{z}_t - \sigma_t \hat{\boldsymbol{\epsilon}}_{\boldsymbol{\theta}}(t, \mathbf{z}_t)}{\alpha_t} + \sigma_{t'} \hat{\boldsymbol{\epsilon}}_{\boldsymbol{\theta}}(t, \mathbf{z}_t), \tag{2}$$

where $\mathbf{z}_{t'}$ will be fed into $\hat{\boldsymbol{\epsilon}}_{\boldsymbol{\theta}}(\cdot)$ again until $t'$ becomes 0, *i.e.*, the denoising process finishes.

**Latent Diffusion Model / Stable Diffusion.** The recent latent diffusion model (LDM) [4] reduces the inference computation and steps by performing the denoising process in the *latent space*, which is encoded from a pre-trained variational autoencoder (VAE) [38, 39]. During inference, the image is constructed through the decoder from the latent. LDM also explores the text-to-image generation, where a text prompt embedding $\mathbf{c}$ is fed into the diffusion model as the condition. When synthesizing images, an important technique, *classifier-free guidance* (CFG) [34], is adopted to improve quality,

$$\tilde{\boldsymbol{\epsilon}}_{\boldsymbol{\theta}}(t, \mathbf{z}_t, \mathbf{c}) = w \hat{\boldsymbol{\epsilon}}_{\boldsymbol{\theta}}(t, \mathbf{z}_t, \mathbf{c}) - (w - 1) \hat{\boldsymbol{\epsilon}}_{\boldsymbol{\theta}}(t, \mathbf{z}_t, \varnothing), \tag{3}$$

where $\hat{\boldsymbol{\epsilon}}_{\boldsymbol{\theta}}(t, \mathbf{z}_t, \varnothing)$ represents the *unconditional* output obtained by using null text $\varnothing$. The guidance scale $w$ can be adjusted to control the strength of conditional information on the generated images to achieve the trade-off between quality and diversity. LDM is further trained on large-scale datasets [40], delivering a series of *Stable Diffusion* (SD) models [4]. We choose Stable Diffusion v1.5 (SD-v1.5) as the baseline. Next, we perform detailed analyses to diagnose the latency bottleneck of SD-v1.5.

## 2.2 Benchmark and Analysis

Here we comprehensively study the parameter and computation intensity of the SD-v1.5. The in-depth analysis helps us understand the bottleneck to deploying text-to-image diffusion models on mobile devices from the scope of network architecture and algorithm paradigms. Meanwhile, the micro-level breakdown of the networks serves as the basis of the architecture redesign and search.

Table 1: **Latency Comparison** between Stable Diffusion v1.5 and our proposed efficient diffusion models (UNet and Image Decoder) on iPhone 14 Pro.

| Stable Diffusion v1.5 | Text Encoder | UNet | VAE Decoder |
|---|---|---|---|
| Input Resolution | 77 tokens | $64 \times 64$ | $64 \times 64$ |
| #Parameters (M) | 123 | 860 | 50 |
| Latency (ms) | 4 | ~1,700[2] | 369 |
| Inference Steps | 2 | 50 | 1 |
| Total Latency (ms) | 8 | 85,000 | 369 |
| **Our Model** | Text Encoder | **Our UNet** | **Our Image Decoder** |
| Input Resolution | 77 tokens | $64 \times 64$ | $64 \times 64$ |
| #Parameters (M) | 123 | **848** | **13** |
| Latency (ms) | 4 | **230** | **116** |
| Inference Steps | 2 | **8** | 1 |
| Total Latency (ms) | 8 | **1,840** | **116** |

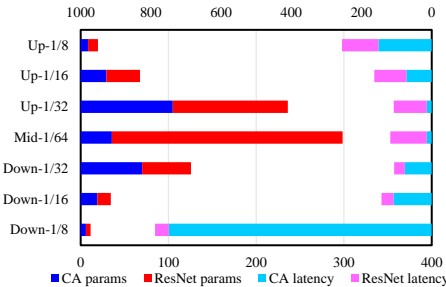

Figure 2: Latency (iPhone 14 Pro, ms) and parameter (M) analysis for cross-attention (CA) and ResNet blocks in the UNet of Stable Diffusion.

**Macro Prospective.** As shown in Tab. 1 and Fig. 3, the networks of stable diffusion consist of three major components. Text encoder employs a ViT-H model [41] for converting input text prompt into embedding and is executed in two steps (with one for CFG) for each image generation process, constituting only a tiny portion of inference latency (8 ms). The VAE decoder takes the latent feature to generate an image, which runs as 369 ms. Unlike the above two models, the denoising UNet is not only intensive in computation (1.7 seconds latency) but also demands iterative forwarding steps to ensure generative quality. For instance, the total denoising timesteps is set to 50 for inference in SD-v1.5, *significantly slowing down* the on-device generation process to the *minute* level.

**Breakdown for UNet.** The time-conditional ($t$) UNet consists of cross-attention and ResNet blocks. Specifically, a cross-attention mechanism is employed at each stage to integrate text embedding ($\mathbf{c}$) into spatial features: $\textit{Cross-Attention}(Q_{\mathbf{z}_t}, K_{\mathbf{c}}, V_{\mathbf{c}}) = \textit{Softmax}(\frac{Q_{\mathbf{z}_t} \cdot K_{\mathbf{c}}^{\top}}{\sqrt{d}}) \cdot V_{\mathbf{c}}$, where $Q$ is projected from noisy data $\mathbf{z}_t$, $K$ and $V$ are projected from text condition, and $d$ is the feature dimension. UNet also uses ResNet blocks to capture locality, and we can formulate the forward of UNet as:

$$\hat{\boldsymbol{\epsilon}}_{\boldsymbol{\theta}}(t, \mathbf{z}_t) = \prod \{\textit{Cross-Attention}(\mathbf{z}_t, \mathbf{c}), \textit{ResNet}(\mathbf{z}_t, t)\}. \tag{4}$$

---

[2]We notice the latency varies depending on the phones and use three phones to get the average speed.

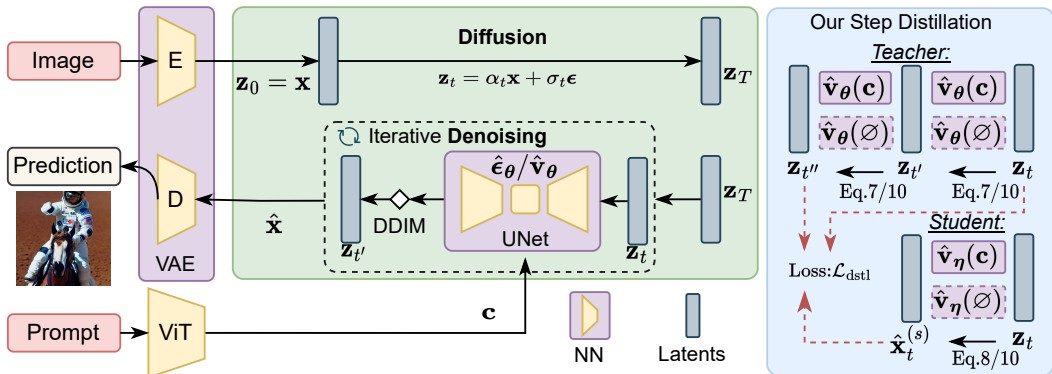

Figure 3: Workflow of text-to-image diffusion model (*left*) and the proposed step distillation (*right*).

The distribution of parameters and computations of UNet is illustrated in Fig. 2, showing that parameters are concentrated on the middle (downsampled) stages because of the expanded channel dimensions, among which ResNet blocks constitute the majority. In contrast, the slowest parts of UNet are the input and output stages with the largest feature resolution, as spatial cross-attentions have quadratic computation complexity with respect to feature size (tokens).

## 3    Architecture Optimizations

Here we investigate the architecture redundancy of SD-v1.5 to obtain efficient neural networks. However, it is non-trivial to apply conventional pruning [42, 43, 44, 45] or architecture search [46, 47, 30] techniques, given the tremendous training cost of SD. Any permutation in architecture may lead to degraded performance that requires fine-tuning with hundreds or thousands of GPUs days. Therefore, we propose an architecture-evolving method that preserves the performance of the pre-trained UNet model while gradually improving its efficacy. As for the deterministic image decoder, we apply tailored compression strategies and a simple yet effective prompt-driven distillation approach.

### 3.1    Efficient UNet

From our empirical observation, the operator changes resulting from network pruning or searching lead to degraded synthesized images, asking for significant training costs to recover the performance. Thus, we propose a robust training, and evaluation and evolving pipeline to alleviate the issue.

**Robust Training.** Inspired by the idea of elastic depth [48, 49], we apply stochastic forward propagation to execute each cross-attention and ResNet block by probability $p(\cdot, I)$, where $I$ refers to identity mapping that skips the corresponding block. Thus, we have Eq. (4) becomes as follows:

$$\hat{\epsilon}_{\boldsymbol{\theta}}(t, \mathbf{z}_t) = \prod \{p(\textit{Cross-Attention}(\mathbf{z}_t, \mathbf{c}), I), p(\textit{ResNet}(\mathbf{z}_t, t), I)\}. \tag{5}$$

With this training augmentation, the network is robust to architecture permutations, which enables an accurate assessment of each block and a stable architectural evolution (more examples in Fig. 5).

**Evaluation and Architecture Evolving.** We perform online network changes of UNet using the model from robust training with the constructed evolution action set: $A \in \{A^{+,-}_{Cross\text{-}Attention[i,j]}, A^{+,-}_{ResNet[i,j]}\}$, where $A^{+,-}$ denotes the action to remove ($-$) or add ($+$) a cross-attention or ResNet block at the corresponding position (stage $i$, block $j$). Each action is evaluated by its impact on execution latency and generative performance. *For latency*, we use the lookup table built in Sec. 2.2 for each possible configuration of cross-attention and ResNet blocks. Note we improve the UNet for on-device speed; the optimization of model size can be performed similarly and is left as future work. *For generative performance*, we choose CLIP score [41] to measure the correlation between generated images and the text condition. We use a small subset (2K images) of MS-COCO validation set [50], fixed steps (50), and CFG scale as 7.5 to benchmark the score, and it takes about 2.5 A100 GPU hours to test each action. For simplicity, the value score of each action is defined as $\frac{\Delta CLIP}{\Delta Latency}$, where a block with lower latency and higher contribution to CLIP tends to be preserved, and the opposite is removed in

architecture evolving (more details in Alg. 1). To further reduce the cost for network optimization, we perform architecture evolving, *i.e.*, removing redundant blocks or adding extra blocks at valuable positions by executing a group of actions at a time. Our training paradigm successfully preserves the performance of pre-trained UNet while tolerating large network permutations (Fig. 5). The details of our final architecture is presented in Sec. A.

## 3.2 Efficient Image Decoder

For the image decoder, we propose a distillation pipeline that uses synthetic data to learn the efficient image decoder obtained via channel reduction, which has $3.8\times$ fewer parameters and is $3.2\times$ faster than the one from SD-v1.5. The efficient image decoder is obtained by applying $50\%$ uniform channel pruning to the original image decoder, resulting in a compressed efficient image decoder with approximately $1/4$ size and MACs of the original one. Here we only train the efficient decoder instead of following the training of VAE [4, 38, 39] that also learns the image encoder. We use text prompts to get the latent representation from the UNet of SD-v1.5 after 50 denoising steps with DDIM and forward it to our efficient image decoder and the one of SD-v1.5 to generate two images. We then optimize the decoder by minimizing the mean squared error between the two images. Using synthetic data for distillation brings the advantage of augmenting the dataset on-the-fly where each prompt be used to obtain unlimited images by sampling various noises. Quantitative analysis of the compressed decoder can be found in Sec. B.2.

---

**Algorithm 1** Optimizing UNet Architecture

---

**Require:** UNet: $\hat{\epsilon}_{\boldsymbol{\theta}}$; validation set: $\mathbb{D}_{val}$; latency lookup table $\mathbb{T} : \{Cross\text{-}Attention[i,j], ResNet[i,j]\}$.
**Ensure:** $\hat{\epsilon}_{\boldsymbol{\theta}}$ converges and satisfies latency objective $S$.
  **while** $\hat{\epsilon}_{\boldsymbol{\theta}}$ not converged **do**
     **Perform robust training**.
     $\rightarrow$ **Architecture optimization**:
     **if** perform architecture evolving at this iteration **then**
        $\rightarrow$ **Evaluate blocks**:
        **for** each $block[i,j]$ **do**
          $\Delta CLIP \leftarrow \text{eval}(\hat{\epsilon}_{\boldsymbol{\theta}}, A^-_{block[i,j]}, \mathbb{D}_{val})$,
          $\Delta Latency \leftarrow \text{eval}(\hat{\epsilon}_{\boldsymbol{\theta}}, A^-_{block[i,j]}, \mathbb{T})$
        **end for**
     $\rightarrow$ **Sort actions based on $\frac{\Delta CLIP}{\Delta Latency}$, execute action, and evolve architecture to get latency $T$:**
        **if** latency objective $S$ is not satisfied **then**
          $\{\hat{A}^-\} \leftarrow \arg\min_{A^-} \frac{\Delta CLIP}{\Delta Latency}$,
        **else**
          $\{\hat{A}^+\} \leftarrow \text{copy}(\arg\max_{A^-} \frac{\Delta CLIP}{\Delta Latency})$,
          $\hat{\epsilon}_{\boldsymbol{\theta}} \leftarrow \text{evolve}(\hat{\epsilon}_{\boldsymbol{\theta}}, \{\hat{A}\})$
        **end if**
     **end if**
  **end while**

---

# 4 Step Distillation

Besides proposing the efficient architecture of the diffusion model, we further consider reducing the number of iterative denoising steps for UNet to achieve more speedup. We follow the research direction of *step distillation* [33], where the inference steps are reduced by distilling the teacher, *e.g.*, at 32 steps, to a student that runs at *fewer* steps, *e.g.*, 16 steps. This way, the student enjoys $2\times$ speedup against the teacher. Here we employ different distillation pipelines and learning objectives from existing works [33, 32] to improve the image quality, which we elaborate on as follows.

## 4.1 Overview of Distillation Pipeline

Citing the wisdom from previous studies [33, 32], step distillation works best with the **v**-prediction type, *i.e.*, UNet outputs *velocity* **v** [33] instead of the noise $\epsilon$. Thus, we fine-tune SD-v1.5 to **v**-prediction (for notation clarity, we use $\hat{\mathbf{v}}_{\boldsymbol{\theta}}$ to mean the SD model in **v**-prediction *vs.* its $\epsilon$-prediction counterpart $\hat{\epsilon}_{\boldsymbol{\theta}}$) before step distillation, with the following original loss $\mathcal{L}_{\text{ori}}$:

$$\mathcal{L}_{\text{ori}} = \mathbb{E}_{t\sim U[0,1],\mathbf{x}\sim p_{\text{data}}(\mathbf{x}),\epsilon\sim\mathcal{N}(\mathbf{0},\mathbf{I})} \, ||\hat{\mathbf{v}}_{\boldsymbol{\theta}}(t, \mathbf{z}_t, \mathbf{c}) - \mathbf{v}||_2^2, \qquad (6)$$

where **v** is the ground-truth target velocity, which can be derived analytically from the clean latent **x** and noise $\epsilon$ given time step $t$: $\mathbf{v} \equiv \alpha_t\epsilon - \sigma_t\mathbf{x}$.

Our distillation pipeline includes three steps. **First**, we do step distillation on SD-v1.5 to obtain the UNet with 16 steps that reaches the performance of the 50-step model. Note here we use a 32-step SD-v1.5 to perform distillation directly, *instead of doing it progressively*, *e.g.*, using a 128-step model as a teacher to obtain the 64-step model and redo the distillation progressively. The reason is that we empirically observe that progressive distillation is slightly worse than direct distillation (see Fig. 6(a)

for details). **Second**, we use the same strategy to get our 16-step efficient UNet. **Finally**, we use the 16-step SD-v1.5 as the teacher to conduct step distillation on the efficient UNet that is initialized from its 16-step counterpart. This will give us the 8-step efficient UNet, which is our final UNet model.

## 4.2 CFG-Aware Step Distillation

We introduce the vanilla step distillation loss first, then elaborate more details on our proposed CFG-aware step distillation (Fig. 3).

**Vanilla Step Distillation.** Given the UNet inputs, time step $t$, noisy latent $\mathbf{z}_t$, and text embedding $\mathbf{c}$, the teacher UNet performs *two* DDIM denoising steps, from time $t$ to $t'$ and then to $t''$ ($0 \leq t'' < t' < t \leq 1$). This process can be formulated as (see the Sec. C for detailed derivations),

$$
\begin{aligned}
\hat{\mathbf{v}}_t = \hat{\mathbf{v}}_{\boldsymbol{\theta}}(t, \mathbf{z}_t, \mathbf{c}) &\Rightarrow \mathbf{z}_{t'} = \alpha_{t'}(\alpha_t \mathbf{z}_t - \sigma_t \hat{\mathbf{v}}_t) + \sigma_{t'}(\sigma_t \mathbf{z}_t + \alpha_t \hat{\mathbf{v}}_t), \\
\hat{\mathbf{v}}_{t'} = \hat{\mathbf{v}}_{\boldsymbol{\theta}}(t', \mathbf{z}_{t'}, \mathbf{c}) &\Rightarrow \mathbf{z}_{t''} = \alpha_{t''}(\alpha_{t'} \mathbf{z}_{t'} - \sigma_{t'} \hat{\mathbf{v}}_{t'}) + \sigma_{t''}(\sigma_{t'} \mathbf{z}_{t'} + \alpha_{t'} \hat{\mathbf{v}}_{t'}).
\end{aligned}
\tag{7}
$$

The student UNet, parameterized by $\boldsymbol{\eta}$, performs only *one* DDIM denoising step,

$$
\hat{\mathbf{v}}_t^{(s)} = \hat{\mathbf{v}}_{\boldsymbol{\eta}}(t, \mathbf{z}_t, \mathbf{c}) \Rightarrow \hat{\mathbf{x}}_t^{(s)} = \alpha_t \mathbf{z}_t - \sigma_t \hat{\mathbf{v}}_t^{(s)},
\tag{8}
$$

where the super-script $(s)$ indicates these variables are for the student UNet. The student UNet is supposed to predict the teacher's noisy latent $\mathbf{z}_{t''}$ from $\mathbf{z}_t$ with just one denoising step. This goal translates to the following vanilla distillation loss objective calculated in the $\mathbf{x}$-space [33, 32],

$$
\mathcal{L}_{\text{vani\_dstl}} = \varpi(\lambda_t) \, || \, \hat{\mathbf{x}}_t^{(s)} - \frac{\mathbf{z}_{t''} - \frac{\sigma_{t''}}{\sigma_t}\mathbf{z}_t}{\alpha_{t''} - \frac{\sigma_{t''}}{\sigma_t}\alpha_t} \, ||_2^2,
\tag{9}
$$

where $\varpi(\lambda_t) = \max(\frac{\alpha_t^2}{\sigma_t^2}, 1)$ is the truncated SNR weighting coefficients [33].

**CFG-Aware Step Distillation.** The above vanilla step distillation can improve the inference speed with no (or only little) FID compromised. However, we do observe the CLIP score turns *obviously worse*. As a remedy, this section introduces a *classifier-free guidance-aware (CFG-aware)* distillation loss objective function, which will be shown to improve the CLIP score significantly.

We propose to perform classifier-free guidance to both the teacher and student before calculating the loss. Specifically, for Eq. (7) and (8), after obtaining the $\mathbf{v}$-prediction output of UNet, we add the CFG step. Take Eq. (8) for an example, $\hat{\mathbf{v}}_t^{(s)}$ is replaced with the following guided version,

$$
\tilde{\mathbf{v}}_t^{(s)} = w\hat{\mathbf{v}}_{\boldsymbol{\eta}}(t, \mathbf{z}_t, \mathbf{c}) - (w-1)\hat{\mathbf{v}}_{\boldsymbol{\eta}}(t, \mathbf{z}_t, \varnothing),
\tag{10}
$$

where $w$ is the CFG scale. In the experiments, $w$ is randomly sampled from a uniform distribution over a range ([2, 14] by default) – this range is called *CFG range*, which will be shown to provide a way to tradeoff FID and CLIP score during training.

After replacing the UNet output with its guided version, all the other procedures remain the same for both the teacher and the student. This gives us a counterpart version of $\mathcal{L}_{\text{vani\_dstl}}$ – which we term *CFG distillation loss*, denoted as $\mathcal{L}_{\text{cfg\_dstl}}$.

**Total Loss Function.** Empirically, we find $\mathcal{L}_{\text{vani\_dstl}}$ helps to achieve low FID while $\mathcal{L}_{\text{cfg\_dstl}}$ helps to achieve high CLIP score (see Fig. 6(c)). To get the best of both worlds, we introduce a *loss mixing* scheme to use the two losses at the same time – A predefined *CFG probability $p$* is introduced, indicating the probability of using the CFG distillation loss in each training iteration (so with $1-p$ probability, the vanilla distillation loss is used). Now, the overall loss can be summarized:

$$
\begin{aligned}
\mathcal{L} &= \mathcal{L}_{\text{dstl}} + \gamma \mathcal{L}_{\text{ori}}, \\
\mathcal{L}_{\text{dstl}} &= \mathcal{L}_{\text{cfg\_dstl}} \text{ if } P \sim U[0,1] < p \text{ else } \mathcal{L}_{\text{vani\_dstl}},
\end{aligned}
\tag{11}
$$

where $\mathcal{L}_{\text{ori}}$ represents the original denoising loss in Eq. (6) and $\gamma$ is its weighting factor; and $U[0,1]$ represents the uniform distribution over range $(0, 1)$.

**Discussion.** As far as we know, only one very recent work [32] studies how to distill the guided diffusion models. They propose to distill CFG into a student model with extra parameters (called $w$-condition) to mimic the behavior of CFG. Thus, the network evaluation cost is reduced by $2\times$ when generating an image. Our proposed solution here is distinct from theirs [32] for at least four

perspectives. **(1)** The general motivations are different. Their $w$-condition model intends to reduce the number of network evaluations of UNet, while ours aims to improve the image quality during distillation. **(2)** The specific proposed techniques are different – they integrate the CFG scale as an input to the UNet, which results in more parameters, while we do not. **(3)** Empirically, $w$-condition model cannot achieve high CLIP scores when the CFG scale is large (as in Fig. 6(b)), while our method is particularly good at generating samples with high CLIP scores. **(4)** Notably, the trade-off of diversity-quality is previously enabled *only during inference* by adjusting the CFG scale, while our scheme now offers a nice property to realize such trade-off *during training* (see Fig. 6(d)), which $w$-condition cannot achieve. This can be very useful for model providers to train different models in favor of quality or diversity.

## 5 Experiment

**Implementation Details.** Our code is developed based on diffusers library[3]. Given step distillation is mostly conducted on **v**-prediction models [33, 32], we fine-tune UNet in our experiments to **v**-prediction. Similar to SD, we train our models on public datasets [51, 40] to report the quantitative results, *i.e.*, FID and CLIP scores (ViT-g/14), on MS-COCO 2014 validation set [50] for zero-shot evaluation, following the common practice [20, 19, 6, 18]. In addition, we collect an internal dataset with high-resolution images to fine-tune our model for more pleasing visual quality. We use 16 or 32 nodes for most of the training. Each node has 8 NVIDIA A100 GPUs with 40GB or 80GB memory. We use AdamW optimizer [52], set weight decay as 0.01, and apply training batch size as 2,048.

Table 2: **Zero-shot** evaluation on MS-COCO 2017 5K subset. Our efficient model is compared against recent arts in the 8-step configuration. Note the compared works use the same model as SD-v1.5, which is much slower than our approach.

| Method | Steps | FID | CLIP |
|---|---|---|---|
| DPM [53] | 8 | 31.7 | 0.32 |
| DPM++ [54] | 8 | 25.6 | 0.32 |
| Meng *et al.* [32] | 8 | 26.9 | 0.30 |
| Ours | 8 | **24.2** | 0.30 |

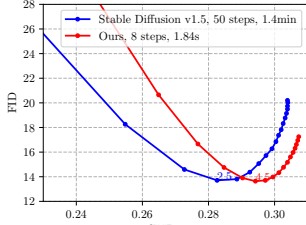 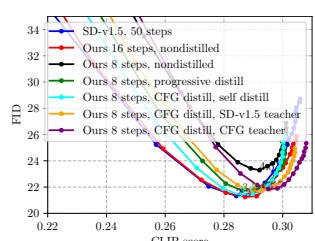

Figure 4: FID *vs.* CLIP on MS-COCO 2014 validation set with CFG scale from 1.0 to 10.0. **Left**: Comparison with SD-v1.5 on *full* set (30K). **Right**: Different settings for step and teacher models tested on 6K samples.

### 5.1 Text-to-Image Generation

We first show the comparison with SD-v1.5 on the *full* MS-COCO 2014 validation set [50] with 30K image-caption pairs. As in Fig. 4 (left), thanks to the architecture improvements and the dedicated loss design for step distillation, our final 8-step, 230ms per step UNet outperforms the original SD-v1.5 in terms of the trade-off between FID *vs.* CLIP. For the most user-preferable guidance scales (ascending part of the curve), our UNet gives about $0.004 - 0.010$ higher CLIP score under the same FID level. In addition, with an aligned sampling schedule (8 DDIM denoising steps), our method also outperforms the very recent distillation work [32] by 2.7 FID with on-par CLIP score, as in Tab. 2. Example synthesized images from our approach are presented in Fig. 1. Our model can generate images from text prompts with high fidelity. More examples are shown in Fig. 9.

We then provide more results for performing step distillation on our efficient UNet. As in Fig. 4 (right), we demonstrate that our 16-step undistilled model provides competitive performance against SD-v1.5. However, we can see a considerable performance drop when the denoising step is reduced to 8. We apply progressive (vanilla) distillation [33, 32] and observe improvements in scores. Though mostly comparable to the SD-v1.5 baseline, the performance of the 8-step model gets saturated for the CLIP score as the guidance scale increases, and is capped at 0.30. Finally, we use the proposed CFG-aware step distillation and find it consistently boosts the CLIP score of the 8-step model with varied configurations. Under the best-observed configuration (CFG distilled 16-step teacher), our 8-step model is able to surpass SD-v1.5 by $0.002 - 0.007$ higher CLIP under similar FID. Discussions on the hyperparameters can be found in ablation studies.

---

[3]https://github.com/huggingface/diffusers

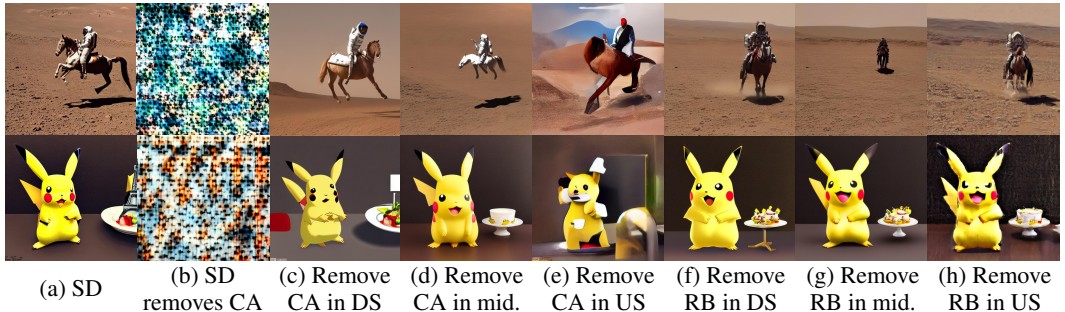

| (a) SD | (b) SD removes CA | (c) Remove CA in DS | (d) Remove CA in mid. | (e) Remove CA in US | (f) Remove RB in DS | (g) Remove RB in mid. | (h) Remove RB in US |

Figure 5: **Advantages of robust training**. Prompts of top row: *a photo of an astronaut riding a horse on mars* and bottom row: *A pikachu fine dining with a view to the Eiffel Tower*. **(a)** Images from SD-v1.5. **(b)** Removing cross-attention (CA) blocks in downsample stage of SD-v1.5. **(c) - (e)** Removing cross-attention (CA) blocks in {downsample (DS), middle (mid.), upsample (US)} using our model after *robust* training. **(f) - (h)** Removing ResNet blocks (RB) in different stages using our model. The model with robust training maintains reasonable performance after dropping blocks.

## 5.2 Ablation Analysis

Here we present the key ablation studies for the proposed approach. For faster evaluation, we test the settings on 6K image-caption pairs randomly sampled from the MS-COCO 2014 validation set [50].

**Robust Training.** As in Fig. 5, we verify the effectiveness of the proposed robust training paradigm. The original model is sensitive to architecture permutations, which makes it difficult to assess the value score of the building blocks (Fig. 5(b)). In contrast, our robust trained model can be evaluated under the actions of architecture evolution, even if multiple blocks are ablated at a time. With the proposed strategy, we preserve the performance of pre-trained SD and save the fine-tuning cost to recover the performance of candidate offspring networks. In addition, we gather some insights into the effect of different building blocks and ensure the architecture permutation is interpretable. Namely, cross-attention is responsible for semantic coherency (Fig. 5(c)-(e)), while ResNet blocks capture local information and are critical to the reconstruction of details (Fig. 5(f)-(h)), especially in the output upsampling stage.

**Step Distillation.** We perform comprehensive comparisons for step distillation discussed in Sec. 4. For the following comparisons, we use the same model as SD-v1.5 to study step distillation.

- Fig. 6(a) presents the comparison of progressive distillation to 8 steps *vs.* direct distillation to 8 steps. As seen, direct distillation wins in terms of both FID and CLIP score. Besides, it is procedurally simpler. Thus, we adopt direct distillation in our proposed algorithm.

- Fig. 6(b) depicts the results of $w$-conditioned models [32] at different inference steps. They are obtained through progressive distillation, *i.e.*, $64 \rightarrow 32 \rightarrow 16 \rightarrow 8$. As seen, there is a clear gap between $w$-conditioned models and the other two, especially in terms of CLIP score. In contrast, our 8-step model can significantly outperform the 50-step SD-v1.5 in terms of CLIP score and maintain a similar FID. Comparing ours (8-step model) to the $w$-conditioned 16-step model, one point of particular note is that, these two schemes have the same inference cost, while ours obviously wins in terms of both FID and CLIP score, suggesting that our method offers a better solution to distilling CFG guided diffusion models.

- Fig. 6(c) shows the effect of our proposed CFG distillation loss *vs.* the vanilla distillation loss. As seen, the vanilla loss achieves the lowest FID, while the CFG loss achieves the highest CLIP score. To get the best of both worlds, the proposed loss mixing scheme (see "vanilla + CFG distill") successfully delivers a better tradeoff: it achieves the similar highest CLIP score as the CFG loss alone *and* the similar lowest FID as the vanilla loss alone.

- There are two hyper-parameters in the proposed CFG distillation loss: CFG range and CFG probability. Fig. 6(d) shows the effect of adjusting them. Only using the vanilla loss (the blue line) and only using the CFG loss (the purple line) lay down two extremes. By adjusting the CFG range and probability, we can effectively find solutions in the middle of the two extremes. As a rule of thumb, *higher* CFG probability and *larger* CFG range will increase the impact of CFG loss, leading to *better* CLIP score but *worse* FID. Actually, for the 7 lines listed top to down in the legend, the

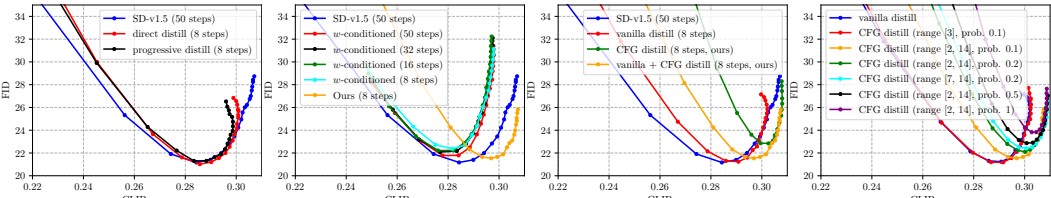

(a) Direct *vs.* progressive  (b) $w$-conditioned *vs.* ours  (c) Vanilla *vs.* CFG distill  (d) CFG hyper-parameters

Figure 6: Ablation studies in step distillation (*best viewed in color*). For each line, from left to right, the CFG scales starts from 1.0 to 10.5 with interval 0.5. **(a)** To obtain the same 8-step student model, in direct distillation, the teacher only distills *once* ($16 \rightarrow 8$), while progressive distillation [33, 32] starts from the 64-step teacher, distills *3 times* to 8 steps ($64 \rightarrow 32 \rightarrow 16 \rightarrow 8$). **(b)** $w$-conditioned model [32] struggles at achieving high CLIP scores (such as over 0.30) while the original SD-v1.5 and our distilled 8-step SD-v1.5 can easily achieve so. **(c)** Comparison between vanilla distillation loss $\mathcal{L}_{\text{vani\_dstl}}$, the proposed CFG distillation loss $\mathcal{L}_{\text{cfg\_dstl}}$, and their mixed version $\mathcal{L}_{\text{dstl}}$. **(d)** Effect of adjusting the two hyper-parameters, CFG range and CFG probability, in CFG distillation. As seen, these hyper-parameters can effectively tradeoff FID and CLIP score.

impact of CFG loss is gradually raised, and we observe the corresponding lines move steadily to the upper right, fully in line with our expectation, suggesting these two hyper-parameters provide a very reliable way to tradeoff FID and CLIP score *during training* – this feature, as far as we know, has not been reported by any previous works.

**Analysis of Original Loss for Distillation.** In Eq. (11), we apply the original denoising loss ($\mathcal{L}_{\text{ori}}$ in Eq. (6)) during the step distillation. Here we show more analysis for the using $\mathcal{L}_{\text{ori}}$ in step distillation.

- Fig. 7(a) shows the comparison between *using* and *not using* the original loss in our proposed CFG distillation method. To our best knowledge, existing step distillation approaches [33, 32] do *not* include the original loss in their total loss objectives, which is actually *sub-optimal*. Our results in Fig. 7(a) suggest that using the original loss can help lower the FID at no loss of CLIP score.

- Fig. 7(b) provides a detailed analysis using different $\gamma$ to balance the original denoising loss and the CFG distillation loss in Eq. (11). We empirically set a dynamic gamma to adjust the original loss into a similar scale to step distillation loss.

**Analysis for the Number of Inference Steps of the Teacher Model.** For the default training setting of the step distillation, the student runs one DDIM step while the teacher runs two steps, *e.g.*, distilling a 16-step teacher to an 8-step student. At the first glance, if the teacher runs more steps, it possibly provides better supervision to the student, *e.g.*, distilling a 32-step teacher to the 8-step student. Here we provide empirical results to show that the approach actually does *not* perform well.

Fig. 7(c) presents the FID and CLIP score plots of different numbers of steps of the teacher model in *vanilla* step distillation. As seen, these teachers achieve similar lowest FID, while the 16-step teacher (blue line) achieves the best CLIP score. A clear pattern is that the more steps of the teacher model, the worse CLIP score of the student. Based on this empirical evidence, we adopt the 16-step teacher setting in our pipeline to get 8-step models.

**Applying Step Distillation to Other Model.** Lastly, we conduct the experiments by applying our proposed CFG-aware distillation on SD-v2, where the student model has the same architecture as SD-v2. The results are provided in Fig. 7(d). As can be seen, our 8-step distilled model achieves comparable performance to the 50-step SD-v2 model. We use the same hyper-parameters from the training of SD-v1.5 for the step distillation of SD-v2, and further tuning might lead to better results.

## 6 Related Work

Recent efforts on text-to-image generation utilize denoising diffusion probabilistic models [55, 35, 56, 1, 2, 4] to improve the synthesis quality by conducting training on the large-scale dataset [40]. However, the deployment of these models requests high-end GPUs for reasonable inference speed due to the tens or hundreds of iterative denoising steps and the huge computation cost of the diffusion

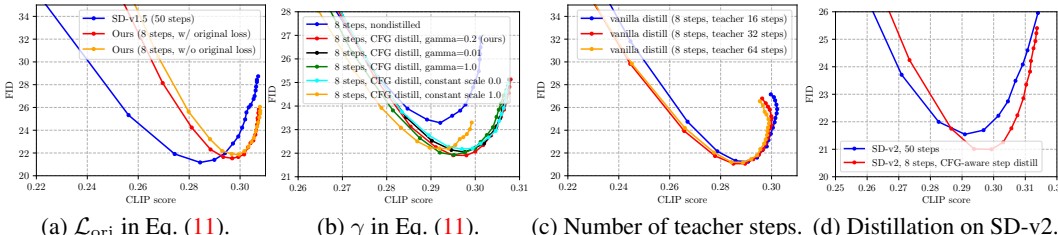

(a) $\mathcal{L}_{\mathrm{ori}}$ in Eq. (11).   (b) $\gamma$ in Eq. (11).   (c) Number of teacher steps.   (d) Distillation on SD-v2.

Figure 7: FID and CLIP results on the 6K samples from the MS-COCO 2014 validation set [50] for various models and experimental settings. **(a)** Comparison between *using (red line)* and *not using (orange line)* the original loss in our proposed CFG distillation method. The hyper-parameter setup of "ours" experiments: CFG range $[2, 14]$ and CFG probability 0.1. **(b)** Analysis of loss scaling $\gamma$ in Eq. (11). Note that we employ dynamic scaling to adjust original loss ($\mathcal{L}_{\mathrm{ori}}$) into a similar scale of step distillation loss ($\mathcal{L}_{\mathrm{dstl}}$). We show $\gamma$ as $0.01, 0.2, 1.0$. Our choice (0.2) gives slightly better FID, despite all dynamic scalings resulting in very similar results. We further show results of constant scaling. Here $0.0$ indicates no $\mathcal{L}_{\mathrm{ori}}$ added, while $1.0$ refers to non-scaled $\mathcal{L}_{\mathrm{ori}}$ where $\mathcal{L}_{\mathrm{ori}}$ dominates the optimization and degrades the effect of step distillation. **(c)** Analysis for the number of steps for the teacher model in vanilla step distillation. The student is supposed to run at $8$ steps, and we can actually employ different teachers that run at different numbers of steps during the step distillation. The default setting in our experiment is that *teacher* 16 *steps, student* 8 *steps*, *i.e.*, the blue line, which turns out to be the best. **(d)** Results of our proposed CFG-aware step distillation applied on SD-v2.

model. This limitation has spurred interest from both the academic community and industry to optimize the efficiency of diffusion models, with two primary approaches being explored: improving the sampling process [57, 58, 59, 60, 53, 61] and investigating on-device solutions [62].

One promising area for reducing the denoising steps is through progressive distillation, where the sampling steps are gradually reduced by distillation that starts from a pre-trained teacher [33]. The later work further improves the inference cost of classifier-free guidance [34] by introducing the $w$-condition [32]. Our work follows the path of step distillation while holding significant differences with existing work, which is discussed above (Sec. 4). Another direction studies the methods for optimizing the model runtime on devices [63], such as post-training quantization [22, 23] and GPU-aware optimization [24]. Nonetheless, these works require specific hardware or compiler support. Our work is orthogonal to post optimizations and can be combined with them for further speed up. We target developing a generic and efficient *network architecture* that can run fast on mobile devices without relying on specific bit width or compiler support. We identify the redundancy in the SD and introduce one with a similar quality while being significantly faster.

# 7   Discussion and Conclusion

This work proposes the fastest on-device text-to-image model that runs denoising in $1.84$ seconds with image quality on par with Stable Diffusion. To build such a model, we propose a series of novel techniques, including analyzing redundancies in the denoising UNet, proposing the evolving-training framework to obtain the efficient UNet model, and improving the step distillation by introducing the CFG-aware distillation loss. We perform extensive experiments and validate that our model can achieve similar or even better quality compared to Stable Diffusion while being significantly faster.

**Limitation.** While our approach is able to run the large-scale text-to-image diffusion model on mobile devices with ultra-fast speed, the model still holds a relatively large number of parameters. Another promising direction is to reduce the model size to make it more compatible with various edge devices. Furthermore, most of our latency analysis is conducted on iPhone 14 Pro, which has more computation power than many other phones. How to optimize our models for other mobile devices to achieve fast inference speed is also an interesting topic to study.

**Broader Impacts.** Similar to existing studies on content generation, our approach must be applied cautiously so that it will not be used for malicious applications. Such concerns can also be alleviated by approaches that could automatically detect image content that violates specific regulations.

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

# A Efficient UNet

We provide the detailed architecture of our efficient UNet in Tab. 3. We perform denoising diffusion in latent space [4]. Consequently, the input and output resolution for UNet is $\frac{H}{8} \times \frac{W}{8}$, which is $64 \times 64$ for generating an image of $512 \times 512$.

In the main paper, we mainly benchmark the latency on iPhone 14 pro. Here we provide the runtine of the model on more mobile devices in Tab. 4.

In addition to mobile phones, we show the latency and memory benchmarks on Nvidia A100 40G GPU, as in Tab. 5. We demonstrate that our efficient UNet achieves over $12\times$ speedup compared to the original SD-v1.5 on a server-level GPU and shrinks $46\%$ running memory. The analysis is performed via the public TensorRT [64] library in single precision.

Table 3: Detailed architecture of our efficient UNet model.

| Stage | Resolution | Type | Config | UNet Model Origin | Ours |
|---|---|---|---|---|---|
| Down-1 | $\frac{H}{8} \times \frac{W}{8}$ | Cross Attention | Dimension | 320 | |
| | | | # Blocks | 2 | 0 |
| | | ResNet | Dimension | 320 | |
| | | | # Blocks | 2 | 2 |
| Down-2 | $\frac{H}{16} \times \frac{W}{16}$ | Cross Attention | Dimension | 640 | |
| | | | # Blocks | 2 | 2 |
| | | ResNet | Dimension | 640 | |
| | | | # Blocks | 2 | 2 |
| Down-3 | $\frac{H}{32} \times \frac{W}{32}$ | Cross Attention | Dimension | 1280 | |
| | | | # Blocks | 2 | 2 |
| | | ResNet | Dimension | 1280 | |
| | | | # Blocks | 2 | 1 |
| Mid | $\frac{H}{64} \times \frac{W}{64}$ | Cross Attention | Dimension | 1280 | |
| | | | # Blocks | 1 | 1 |
| | | ResNet | Dimension | 1280 | |
| | | | # Blocks | 7 | 4 |
| Up-1 | $\frac{H}{32} \times \frac{W}{32}$ | Cross Attention | Dimension | 1280 | |
| | | | # Blocks | 3 | 6 |
| | | ResNet | Dimension | 1280 | |
| | | | # Blocks | 3 | 2 |
| Up-2 | $\frac{H}{16} \times \frac{W}{16}$ | Cross Attention | Dimension | 640 | |
| | | | # Blocks | 3 | 3 |
| | | ResNet | Dimension | 640 | |
| | | | # Blocks | 3 | 3 |
| Up-3 | $\frac{H}{8} \times \frac{W}{8}$ | Cross Attention | Dimension | 320 | |
| | | | # Blocks | 3 | 0 |
| | | ResNet | Dimension | 320 | |
| | | | # Blocks | 3 | 3 |

Table 4: Latency benchmark on iPhone12 Pro Max, iPhone13 Pro Max, and iPhone 14 Pro.

| Device | Text Encoder (ms) | UNet (ms) | VAE Decoder (ms) | Overall (s) |
|---|---|---|---|---|
| iPhone14 Pro | 4.0 | 230 | 116 | 1.96 |
| iPhone13 Pro Max | 5.7 | 315 | 148 | 2.67 |
| iPhone12 Pro Max | 6.3 | 526 | 187 | 4.40 |

Table 5: Latency analysis on Nvidia A100 40G GPU with the TensorRT [64] library, tested with single precision (FP32).

| UNet | Batch Size | Latency (ms) | Memory (MB) | Iters | Total Latency (ms) | Speedup |
|---|---|---|---|---|---|---|
| SD-v1.5 | 2 | 51.2 | 6634 | 50 | 2,560 | - |
| Ours | 2 | **26.2** | **3549** | **8** | **209.6** | **12.2×** |

## B Discussions of Text Encoder and VAE Decoder

### B.1 Text Encoder

Exiting works have explored the importance of the pre-trained text encoder for generating images [19, 20]. In our work, considering the negligible inference latency (4ms) of the text encoder compared to the UNet and VAE Decoder, we do not compress the text encoder in the released pipeline.

### B.2 VAE Decoder

We provide qualitative visualizations and quantitive results of our compressed VAE decoder in Fig. 8. The main paper shows that the image decoder constitutes a small portion of inference latency (369ms) compared to the original UNet from SD-v1.5. However, regarding our optimized pipeline (230ms × 8 steps), the decoder consumes a considerable portion of overall latency. We propose an effective distillation paradigm to compress the VAE decoder. Specifically, we obtain the latent-image pairs by forwarding the text prompts into the original SD-v1.5 model. The student, which is the compressed decoder, takes the latent from the teacher model as input and generates an output image that is optimized with the ones from the teacher model by the mean squared error. Our proposed method wields the following advantages. First, our approach does not demand paired text-image samples, and it can generate unlimited data on-they-fly, benefiting the generalization of the compressed decoder. Second, the distillation paradigm is simple and straightforward, requiring minimal implementation efforts compared to conventional VAE training. As in Fig. 8, our compressed decoder (116ms) provides comparable generative quality, and the performance degradation compared to the original VAE decoder is negligible.

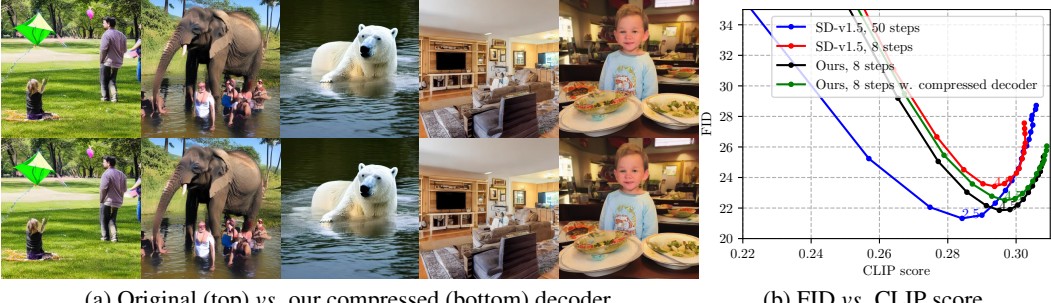

(a) Original (top) *vs.* our compressed (bottom) decoder.         (b) FID *vs.* CLIP score.

Figure 8: Evaluation using MS-COCO 2014 validation set [50]. **(a)** Generated images by using the decoder from SD-v1.5 and our compressed image decoder. The UNet is our efficient UNet, and the guidance scale for CFG is 9.0. **(b)** Quantitative comparison on the 6K samples. Our compressed decoder performs similarly to the original one considering the widely used CFG scale, *i.e.*, from 7 to 9, and still performs better than the SD-v1.5.

## C Detailed Derivations of Step Distillation

The following are the detailed derivations of Eq. (7) ∼ Eq. (9) in the main paper.

Given the UNet inputs, time step $t$, noisy latent $\mathbf{z}_t$, and text embedding $\mathbf{c}$, the teacher UNet performs *two* DDIM denoising steps, from time $t$ to $t'$ and then to $t''$ ($0 \leq t'' < t' < t \leq 1$).

We first examine the process from $t$ to $t'$, which can be formulated as,

$$
\begin{aligned}
\hat{\mathbf{v}}_t &= \hat{\mathbf{v}}_{\boldsymbol{\theta}}(t, \mathbf{z}_t, \mathbf{c}) \quad \triangleright \texttt{Teacher UNet first forward} \\
\Rightarrow \hat{\mathbf{x}}_t &= \alpha_t \mathbf{z}_t - \sigma_t \hat{\mathbf{v}}_t, \quad \triangleright \texttt{Teacher predicted clean latent at time } t \\
\hat{\boldsymbol{\epsilon}}_t &= \sigma_t \mathbf{z}_t + \alpha_t \hat{\mathbf{v}}_t, \quad \triangleright \texttt{Teacher predicted noise at time } t \\
\Rightarrow \mathbf{z}_{t'} &= \alpha_{t'} \hat{\mathbf{x}}_t + \sigma_{t'} \hat{\boldsymbol{\epsilon}}_t \quad \triangleright \texttt{Teacher predicted noisy latent at time } t' \\
&= \alpha_{t'}(\alpha_t \mathbf{z}_t - \sigma_t \hat{\mathbf{v}}_t) + \sigma_{t'}(\sigma_t \mathbf{z}_t + \alpha_t \hat{\mathbf{v}}_t).
\end{aligned}
\tag{12}
$$

The process from $t'$ to $t''$ can be derived just like the above, by replacing $t$ and $t'$ with $t'$ and $t''$, respectively:

$$\hat{\mathbf{v}}_{t'} = \hat{\mathbf{v}}_{\boldsymbol{\theta}}(t', \mathbf{z}_{t'}, \mathbf{c}) \quad \triangleright \texttt{Teacher UNet second forward}$$

$$\Rightarrow \hat{\mathbf{x}}_{t'} = \alpha_{t'}\mathbf{z}_{t'} - \sigma_{t'}\hat{\mathbf{v}}_{t'}, \quad \triangleright \texttt{Teacher predicted clean latent at time } t'$$

$$\hat{\boldsymbol{\epsilon}}_{t'} = \sigma_{t'}\mathbf{z}_{t'} + \alpha_{t'}\hat{\mathbf{v}}_{t'}, \quad \triangleright \texttt{Teacher predicted noise at time } t' \tag{13}$$

$$\Rightarrow \mathbf{z}_{t''} = \alpha_{t''}\hat{\mathbf{x}}_{t'} + \sigma_{t''}\hat{\boldsymbol{\epsilon}}_{t'} \quad \triangleright \texttt{Teacher predicted noisy latent at time } t''$$

$$= \alpha_{t''}(\alpha_{t'}\mathbf{z}_{t'} - \sigma_{t'}\hat{\mathbf{v}}_{t'}) + \sigma_{t''}(\sigma_t\mathbf{z}_{t'} + \alpha_t\hat{\mathbf{v}}_{t'}).$$

The student UNet, parameterized by $\boldsymbol{\eta}$, performs only *one* DDIM denoising step,

$$\hat{\mathbf{v}}_t^{(s)} = \hat{\mathbf{v}}_{\boldsymbol{\eta}}(t, \mathbf{z}_t, \mathbf{c}) \quad \triangleright \texttt{Student UNet forward}$$

$$\Rightarrow \hat{\mathbf{x}}_t^{(s)} = \alpha_t\mathbf{z}_t - \sigma_t\hat{\mathbf{v}}_t^{(s)}, \quad \triangleright \texttt{Student predicted clean latent at time } t$$

$$\hat{\boldsymbol{\epsilon}}_t^{(s)} = (\mathbf{z}_t - \alpha_t\hat{\mathbf{x}}_t^{(s)})/\sigma_t, \quad \triangleright \texttt{Student predicted noise at time } t$$

$$\Rightarrow \mathbf{z}_{t''}^{(s)} = \alpha_{t''}\hat{\mathbf{x}}_t^{(s)} + \sigma_{t''}\hat{\boldsymbol{\epsilon}}_t^{(s)} \quad \triangleright \texttt{Student predicted noisy latent at time } t''$$

$$= \alpha_{t''}\hat{\mathbf{x}}_t^{(s)} + \frac{\sigma_{t''}}{\sigma_t}(\mathbf{z}_t - \alpha_t\hat{\mathbf{x}}_t^{(s)}) \tag{14}$$

$$= (\alpha_{t''} - \frac{\sigma_{t''}}{\sigma_t}\alpha_t)\hat{\mathbf{x}}_t^{(s)} + \frac{\sigma_{t''}}{\sigma_t}\mathbf{z}_t,$$

$$\Rightarrow \hat{\mathbf{x}}_t^{(s)} = \frac{\mathbf{z}_{t''}^{(s)} - \frac{\sigma_{t''}}{\sigma_t}\mathbf{z}_t}{\alpha_{t''} - \frac{\sigma_{t''}}{\sigma_t}\alpha_t},$$

where the super-script $(s)$ indicates these variables are for the student UNet. The student UNet is supposed to predict the noisy latent $\mathbf{z}_{t''}$ from $\mathbf{z}_t$ of the teacher with just one denoising step, namely,

$$\mathbf{z}_{t''}^{(s)} = \mathbf{z}_{t''}. \tag{15}$$

Replacing $\mathbf{z}_{t''}^{(s)}$ with $\mathbf{z}_{t''}$ in the final equation of Eq. (14), we arrive at the following loss objective,

$$\mathcal{L}_{\text{vani\_dstl}} = \varpi(\lambda_t) \, || \, \hat{\mathbf{x}}_t^{(s)} - \frac{\mathbf{z}_{t''} - \frac{\sigma_{t''}}{\sigma_t}\mathbf{z}_t}{\alpha_{t''} - \frac{\sigma_{t''}}{\sigma_t}\alpha_t} \, ||_2^2, \tag{16}$$

where $\varpi(\lambda_t) = \max(\frac{\alpha_t^2}{\sigma_t^2}, 1)$ is the truncated SNR weighting coefficients [33].

## D   Different Teacher Options for Step Distillation

It is non-trivial to decide the best teacher model to distill our final 8-step efficient UNet. In Fig. 4, we conduct several experiments to explore different teacher options. As straightforward choices, self-distillation from our 16-step efficient UNet or distillation from the 16-step SD-v1.5 baseline model can effectively boost the performance of our 8-step model. Additionally, we investigate whether stronger teachers can further boost performance by training a CFG-aware distilled 16-step SD-v1.5 model, as discussed in Sec. 4. We obtain significant improvements in CLIP scores, demonstrating the potential of employing better teacher models. We would like to mention that we also experiment with SD-v2 as the teacher model. Surprisingly, we observe much worse results. We attribute this to the different text embeddings used in SD-v1.5 and SD-v2 pipelines. Distillation between different infrastructures might be a possible future direction to explore.

## E   Additional Qualitative Results

We provide more generated images from our text-to-image diffusion model in Fig. 9. As an acceleration work for generic Stable Diffusion [4], our efficient model demonstrates a sufficient capability to synthesize various contents with high aesthetics, such as realistic objects (food, animals), scenery, and artistic and cartoon styles.

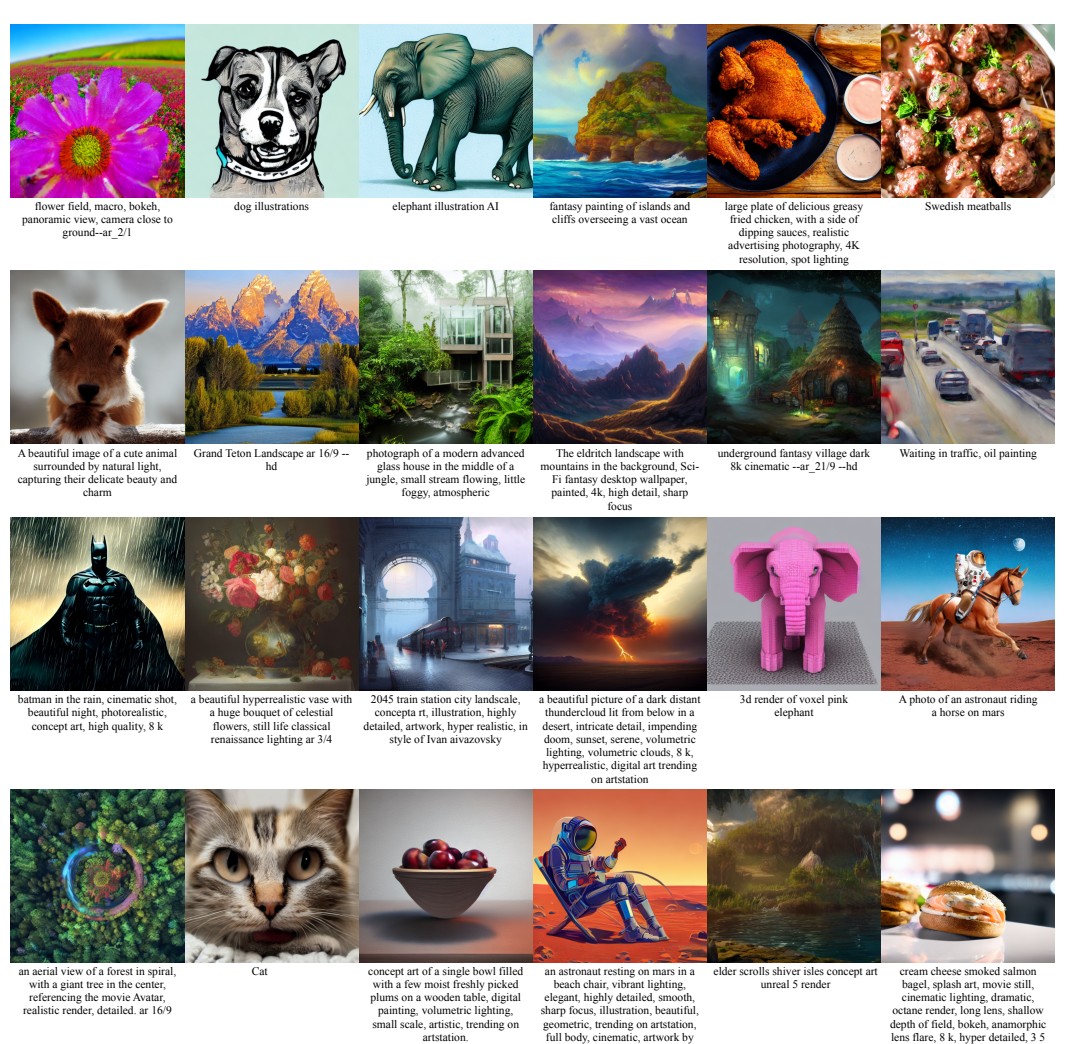

Figure 9: Example generated images by using our efficient text-to-image diffusion model.

