# OpenReview forum: "SnapFusion: Text-to-Image Diffusion Model on Mobile Devices within Two Seconds"
_NeurIPS.cc/2023/Conference — NeurIPS 2023 poster_

### Official Review · Reviewer_oXj2 · 2023-06-12

**Soundness:** 3 good
**Presentation:** 4 excellent
**Contribution:** 2 fair
**Rating:** 5
**Confidence:** 4

**Summary:**

This paper propose a system of model compression methods for text-to-image models such as stable diffusion. Authors propose to apply robust training (i.e. stochastic depth training) to train a network robust to architecture change and propose CFG-free knowledge distillation during the compression of the u-net. Besides, author also give in-depth study on the computation cost of different layers.

**Strengths:**

1. This paper focus on a hot and an interesting topic - compression of large generative models.
2. This paper have good evalution and good experimental results.
3. Good writing.
3. I like the video in the supplementary material.

**Weaknesses:**

1. The main model compression methods are borrowed from previous methods. Robust training is totally same as stochastic depth training. Their evaluation methods for architecture search is also well used in NAS. In the knowledge distillation part, simply adding knowledge distillation loss CFG guidance does not show enough novelty.
2. About the experimental results. Their have been many KD methods for stable diffusion (e.g. On Distillation of Guided Diffusion Models). No enough comparison is provided. Besides, although the appendix gives some qualitative results, it still will be better to provide more.
3. One of the main obstacles in compression of large-scale generative models is the overlarge training cost. Although authors provide their training settings in the manuscript,  it will be better to provide more information about the training costs of robust training and the architecture optimization.

**Questions:**

Please refer to the weakness.

---

> ### Author Rebuttal · Authors · 2023-08-09
>
> **We thank the reviewer for the positive feedback and valuable comments. We appreciate that the reviewer acknowledges our paper studies a hot and interesting topic, and includes good evaluation and experimental results. We are glad to know the reviewer likes the video in the supplementary material. We address the raised concerns as follows.**
>
> ***
> **Q1. About novelty.**
>
> We humbly think our proposed architectural evolving and CFG distillation are both novel. First, we agree that the architectural evolving is inspired by the elastic depth, which we have cited on Line 117. However, to the best of our knowledge, we are the first to study how to efficiently optimize the architecture of large-scale text-to-image diffusion models, which is rarely studied before. The process requires careful design and evaluation metrics. Second, we comprehensively study the effects of step distillation and propose a new CFG distillation that can better trade off the FID/CLIP score than existing works. Therefore, we think our proposed approaches are valuable and novel, which is also agreed by Review 5sgC, Reviewer vQqa, and Reviewer 3k2m.
>
> ***
> **Q2. About comparisons to KD methods for stable diffusion, and more qualitative results.**
>
> We carefully reviewed the related literature and there are two works [a,b] that are highly related to our step distillation. The two works use progressive step distillation [a, b], and [b] proposes the w-conditioned model. In the paper, we have comprehensively compared our results with previous works [a,b]. For example, in Fig. 6(a), we compare our direction distillation with progressive distillation. In Fig. 6 (b), we compare our results with the w-conditioning method proposed in [b]. The extensive comparison demonstrates that the strategy for our step distillation is better than existing works. Additionally, in Tab. 2 of the main paper, we further compare our efficient UNet with the model in [b], which has a similar architecture as SD-v1.5, and demonstrate that our model can achieve better FID under the same DDIM scheduler. Additionally, we notice that in Fig.10 in [b], the distilled model is capped with less than 0.30 CLIP score, showing degraded text-image alignment, while our 8-step model overcomes this issue and achieves even better CLIP than SD-v1.5 (approaching 0.31).
>
> For the qualitative results, we provide images in Fig. 1 of the main paper, Fig. 7 and Fig. 10 in the supplementary file, and in the demo video. We further provide more generated images in the author response PDF (Fig. 3).
>
> ***
> **Q3. About training cost.**
>
> Thanks for the question about training costs. We provide more details in the following table, including the cost for robust training, efficient UNet fine-tuning, step distillation, and decoder compression. Overall, our model training only requires 8.6% training samples in comparison with the training of SD-v1.5 from scratch. Note here we report training iterations and total training samples because the GPU hour estimation is dependent on GPU cluster specs, especially inter-node bandwidth, and thus inaccurate. Based on public information (by Huggingface), SD-v1.5 is trained with 30-60 days on 32 nodes, while replicating our entire workflow takes about 4 days on our 32-node cluster.
>
>
> >|           Stage           | A100 GPUs | Batch size | 256x256 iters (K) | 512x512 iters (K) | Total training samples (M) |
> >|-------------------------|:---------:|:----------:|:-----------------:|:-----------------:|:--------------------------:|
> >|      Robust training      |    128    |      4     |         -         |         25        |            12.8            |
> >| Efficient UNet finetuning |    256    |      8     |         -         |         55        |            112.6           |
> >|     Step distillation     |    128    |      4     |         -         |         10        |             5.1            |
> >|    Decoder compression    |       96    |    6        |         -         |         10          |           5.7                 |
> >|   SD-v1.5 (from scratch)  |    256    |      4     |        237        |        1304       |           1578.0           |
>
> ***
> References:
>
> [a] Salimans, Tim, et al., Progressive distillation for fast sampling of diffusion models. ICLR, 2022.
>
> [b] Meng, Chenlin, et al., On distillation of guided diffusion models. CVPR, 2023.

---

> > ### Author Response · Authors · 2023-08-20
> >
> > Dear Reviewer oXj2,
> >
> > Thank you for your valuable feedback and positive rating.
> >
> > We provide additional explanations to help clarify our work. As the deadline for Author-Reviewer discussion is approaching, we would like to use this opportunity to see if our responses are sufficient and if any concern remains. Thanks again for your time.
> >
> >
> > Best,
> >
> > Authors

---

### Official Review · Reviewer_3k2m · 2023-07-05

**Soundness:** 3 good
**Presentation:** 4 excellent
**Contribution:** 3 good
**Rating:** 7
**Confidence:** 5

**Summary:**

This work develops a lightweight Stable Diffusion model with architecture compression and step reduction. For architectural compression, an efficient UNet architecture is obtained from evaluating the importance of individual residual and attention blocks, and an efficient image decoder is obtained via channel reduction and conventional knowledge distillation. For step reduction, a distillation that applies classifier-free guidance (CFG) during the training phase is introduced. Unlike previous methods that apply CGF during the inference, the proposed approach achieves a better tradeoff between FID and CLIP score.


**Strengths:**

* This work successfully compresses Stable Diffusion, one of the most famous foundation models, and enables their deployment on mobile devices with a two-second latency. I think this study can receive significant attention in both academia and industry fields.
* The results of the proposed architectural compression are impressive, although some questions regarding training computations and resources exist (described below).
* The improved step distillation is well-motivated and novel. I think this can be broadly applicable to diffusion-based large models beyond Stable Diffusion.
* I appreciate Figure 2 and Table 1, which effectively explain the compute cost of SD. In particular, Figure 2 is very informative and well illustrates architectural bottlenecks in parameters (inner stages) and latency (outer stages).
* The paper is very well-written and easy-to-follow. The terms and equations are properly described.


**Weaknesses:**

* According to the implementation details, this work seems to use the following datasets and compute machines. The training cost feels VERY HUGE, in contrast to the authors’ initial claim at the introductory paragraph of Section 3 (i.e., pruning and architecture search require significant training compute, whereas the authors propose some methods to alleviate the issue). It seems like a training cost comparable to building the original SD model from scratch.
    * [49] Coyo-700m: Image-text pair dataset. https://github.com/kakaobrain/coyo-dataset, 2022.
    * [38] Laion-5b: An open large-scale dataset for training next generation image-text models. arXiv preprint arXiv:2210.08402, 2022.
    * [+] an internal dataset with high-resolution images to fine-tune our model for more pleasing visual quality.
    *  At least, 16 nodes x 8 GPUs per node = 128 A100 GPUs (We use 16 or 32 nodes for most of the training. Each node has 8 NVIDIA A100 GPUs with 40GB or 80GB memory)

* Would you clarify the number of training pairs AND the training hours used for each of (1) Robust Training in Section 3.1, (2) Image Decoder Compression in Section 3.2, and (3) Step Distillation in Section 4? I think the detailed description of training data/hours is quite important for readers and future research, especially under the popularity of large foundation models.

* Furthermore, are there any supporting materials or results to argue “From our empirical observation, the operator changes resulting from network pruning or searching lead to degraded synthesized images, asking for significant training costs to recover the performance.” at the introductory paragraph of Section 3? A recent, concurrent work* claims that it would be possible to obtain a small pruned UNet in SD with very small retraining cost.
    * On Architectural Compression of Text-to-Image Diffusion Models, https://arxiv.org/abs/2305.15798

* Is there any overlap between the data used for architecture evolving (a smaller subset (2K images) of MS-COCO validation set [48] in line 130 at page 4) and the zero-shot MS-COCO set for final evaluation? If yes, it cannot be considered as the zero-shot evaluation, because the development data for selecting proper blocks were already used and seen. Furthermore, I was wondering the type and quantity of data for architecture evolving would matter and affect the performance.

* Robust Training in Section 3.1 - Would you clarify whether random block dropping is applied to training from scratch (i.e., random initialization) or to retraining from the pretrained SD-v1.5?

* It would be better to describe the method details about compressing the image decoder. Could you describe the channel reduction and the type of synthetic data in detail? I was not able to find them although I have checked the supplementary material (e.g., Supple Section 2.2 VAE Decoder).

* It would be good to analyze the impact of gamma in Eqn (11), the loss weighting between the vanilla step distillation loss and the proposed CFG-aware distillation loss. How did the authors set this parameter? How did different gamma values affect the generation performance? I have checked Figure (6)-b to analyze the effect of CFG range of w in Eqn (10) and the CFG probability p of the loss mixing in Eqn (11), but the effect of gamma in Eqn (11) is not explained.


**Questions:**

Please check the above weakness section.

---

> ### Author Rebuttal · Authors · 2023-08-09
>
> **We thank the reviewer for the positive feedback and thoughtful comments. We appreciate that the reviewer acknowledges that our successfully compressed two-second text-to-image model can achieve significant attention in academia and industry, our step distillation is novel and can be broadly applied to models beyond Stable Diffusion, our results are informative and impressive, and our paper is well-written.**
>
> ***
> **Q1. Training dataset and detailed training cost per each stage.**
>
> Thanks for raising the question. In our training pipelines, we use 189M text-image pairs from Laion-5b with the shortest edge as 1024, and 194M text-image pairs from Coyo-700M with the shortest edge as 512.  As mentioned in Lines 230-232, we report all the quantitative experiments in the paper by only using the above public datasets for fair comparisons with existing works. The data we used only accounts for 6.8% of the data in Laion-5b and Coyo-700M.
>
> Some qualitative images, such as Fig. 1 in paper, are generated by using the model fine-tuned on additional dataset, as in Lines 232-233, which has 160M text-image pairs.  Fine-tuning our model on the additional dataset is optional and achieves very similar quantitative results as public datasets, while sometimes generating more visually-pleasing images.
>
> We provide detailed training costs in the following table, including the cost of robust training, efficient UNet fine-tuning, step distillation, and decoder compression. Overall, our model training only requires 8.6% training samples in comparison with the training of SD-v1.5 from scratch. Note here we report training iterations and total training samples because the GPU-hour estimation is highly dependent on GPU cluster specs, especially inter-node bandwidth, and thus inaccurate.  Based on public information (Huggingface), SD-v1.5 is trained with 30-60 days on 32 nodes (8 GPUs per node), while replicating our entire workflow takes $\sim 4$ days on our 32-node cluster.
>
>
> >|           Stage           | A100 GPUs | Batch size | 256x256 iters (K) | 512x512 iters (K) | Total training samples (M) |
> >|-------------------------|:---------:|:----------:|:-----------------:|:-----------------:|:--------------------------:|
> >|      Robust training      |    128    |      4     |         -         |         25        |            12.8            |
> >| Efficient UNet finetuning |    256    |      8     |         -         |         55        |            112.6           |
> >|     Step distillation     |    128    |      4     |         -         |         10        |             5.1            |
> >|    Decoder compression    |       96    |    6        |         -         |         10          |           5.7                 |
> >|   SD-v1.5 (from scratch)  |    256    |      4     |        237        |        1304       |           1578.0           |
>
> ***
> **Q2. Discussions on pruning cost and concurrent work.**
>
> We tried to apply simple channel pruning on UNet. Under similar training time as robust training and efficient UNet fine-tuning (more than 80K iterations), the model is still hard to recover ($\sim 0.02$ drop in CLIP score). More advanced channel pruning methods are required.
>
> Thanks for mentioning the concurrent work [a], which is an interesting and inspiring work! We notice three major differences between our work and [a] that might cause the differences in training cost. First, we do step distillation to reduce the inference time, while [a] does not. Second, we use the automatic approach for finding and adding blocks, while [a] uses a manually designed pipeline. The cost of manual labor is not clear. Third, and most importantly, we aim to achieve a similar or even better FID/CLIP score than the Stable Diffusion models, while [a] performs slightly worse than SD baseline (FID degrades by $2.71\sim 4.07$, CLIP degrades by $0.008\sim 0.03$). We will cite and discuss [a] in the revised paper.
>
> ***
> **Q3. About the COCO 2K validation data for architecture evolving.**
>
> There is no overlap between the 2K data for architecture validation and the data used for our evaluation on MS-COCO (6K or 30K).
>
> As suggested, we show the results of using different data, i.e., 2K subset from MS-COCO and 2K subset from LAION, to validate actions in architecture evolving. As shown in the response PDF (Fig.2.a), using different data leads to very similar conclusions ($\Delta  CLIP$) for each block. We show stability regarding the validation set selection and strong correlations of the block importance estimation. For instance, both COCO and LAION validation sets suggest the importance of Up.2.Attention, i.e., we should apply more attention modules in Up.2 stage.
>
> ***
> **Q4. About robust training.**
>
> Robust training in Section 3.1 is applied to pre-trained SD-v1.5.
>
> ***
> **Q5. Compression details of image decoder.**
>
> We apply 50% uniform channel pruning to the image decoder, resulting in a compressed efficient image decoder with approximately $\frac{1}{4}$ size and MACs.
>
> In order to train the efficient image decoder, we use the dataset that only includes the text prompts from the LAION dataset. For each text prompt, we get the latent representation from the UNet of SD-v1.5 after 50 denoising steps with DDIM. We then forward the latent to our efficient image decoder and the decoder from SD-v1.5, to generate two outputs. Then we distill the efficient decoder by minimizing the mean squared error between the two outputs. We provide the details on Line 148 and will improve it with more details in the revision.
>
> ***
> **Q6. About gamma in Eq.11.**
>
> Thank you for your valuable comment. We empirically set a dynamic gamma to adjust the original loss into a similar scale to step distillation loss. We provide a detailed ablation analysis of different scaling strategies in the response PDF (Fig.2.b) and will include it in the revision.
>
> ***
> References:
>
> [a] Kim, et al., On Architectural Compression of Text-to-Image Diffusion Models, 2023.

---

> > ### Comment · Reviewer_3k2m · 2023-08-16
> > **Post-rebuttal review**
> >
> > Dear Authors,
> >
> > I sincerely appreciate the time and effort for the rebuttal and would like to increase my score from 6 to 7 for the following reasons:
> > - The authors have thoroughly described the training computational cost. While a concern remains regarding whether many researchers can access to such computing resources and can reproduce this work, it should not be a determining factor in evaluating this work. I also deeply appreciate the detailed discussions on a channel pruning baseline and a concurrent study.
> > - The authors ensured that the evaluation results were genuinely zero-shot and conducted further examinations regarding the impact of different data on architecture evolution. Additionally, the choice of the hyperparameter gamma was investigated during this rebuttal. Thanks for the clarification and additional experiments.
> > - The starting point of robust training, i.e., pretrained SD-v1.5 (not trained from scratch), is understandable, and the details of compressing the image decoder are addressed.
> > - The newly conducted experiments on SD-v2 look impressive, and I believe they can attract attention from the community.

---

> > > ### Author Response · Authors · 2023-08-17
> > > **Thank you for the response**
> > >
> > > Dear Reviewer 3k2m,
> > >
> > > Thank you for your positive response! We are glad to know that your questions have been answered. We will include the additional discussions and analysis in our revised paper.
> > >
> > > Best regards,
> > >
> > > Authors

---

### Official Review · Reviewer_vQqa · 2023-07-07

**Soundness:** 2 fair
**Presentation:** 2 fair
**Contribution:** 3 good
**Rating:** 7
**Confidence:** 3

**Summary:**

This paper introduces a novel framework that significantly reduces the inference speed of text-to-image diffusion models to under two seconds on mobile devices. The authors propose two key techniques. Firstly, they introduce Efficient U-Net which enhances inference speed by eliminating redundant architectural components without compromising performance. Secondly, the authors present a novel step distillation method that incorporates a classifier-free guidance loss term, which further improves CLIP score. Extensive experiments are conducted on the MS-COCO dataset to validate the effectiveness of the proposed approach.


**Strengths:**

- The paper tackles a timely and practically-relevant problem supported by a fair amount of experiments, and stands as the pioneering study in attempting to reduce the latency of text-to-diffusion models on mobile devices.
- The paper covers a good amount of relevant previous studies.

**Weaknesses:**

- In lines 232-233, the authors mention the use of an additional dataset for the fine-tuning purpose. It is crucial for the authors to provide further clarification regarding the fair comparison issue in relation to this additional dataset.
- The proposed distillation scheme appears to involve a series of intricate choices, making it less readily applicable to other settings. For instance, can this distillation scheme be applied to SD-v2.0?
- The clarity of Algorithm 1 should be improved. For instance, the condition “T is not satisfied” and $\hat{A}$ needs clarification. Also, please write the details regarding removing a group of actions in line 136.
- The readability of Section 4.1 can be enhanced, particularly by providing an explicit distillation algorithm. Further, the reason as to how step distillation and cfg loss trade off FID/CLIP score is unclear.
- While the image generation speed has been enhanced, the model still carries a memory burden.

I am willing to raise my score if above concerns are properly addressed during rebuttal.

**Questions:**

- In Fig 5, can authors provide the generated images under SD without ResBlocks (before robust training)?
- Just out of curiosity, did authors conduct experiments on transformer-based diffusion models (e.g., DiT [1]) as well?

[1] Peebles, William, and Saining Xie. (2022) "Scalable diffusion models with transformers."

**Limitations:**

The paper provided limitations along with future research directions in Section 7.

---

> ### Author Rebuttal · Authors · 2023-08-09
>
> **We thank the reviewer for the positive feedback and valuable suggestions. We appreciate that the reviewer acknowledges our paper introduces a novel framework with efficient Unet and novel step distillation to significantly reduce the inference speed of the text-to-image models, and stands as the pioneering study. We also thank the reviewer for agreeing the problem studied in this paper is timely and practically relevant with extensive experiments. We address the main questions in the following.**
>
> ***
> **Q1. About the additional dataset.**
>
> We report all the quantitative experiments in the paper by only using public datasets for fair comparisons with existing works, as mentioned in Lines 230-232.
>
> Some qualitative images in the paper, such as Fig. 1 in the main paper, are generated by using the model fine-tuned on additional dataset, as mentioned on Lines 232-233. Fine-tuning our model on the additional dataset is optional and actually achieves very similar quantitative results as using public datasets, while sometimes generating more visually-pleasing images.
>
> ***
> **Q2. Distillation scheme on Stable Diffusion V2.**
>
> Thanks for the suggestion. Our distillation pipeline is a generic approach without relying on specific models. We conduct the distillation experiments on SD-v2 and plot the FID and CLIP scores in the author response PDF (Fig.1.a). As we can see, our 8-step distilled model achieves comparable performance to the 50-step SD-v2 model. Please note that we use the exact same hyper-parameters from the training of SD-v1.5 for the step distillation of SD-v2, and further tuning might lead to better results.
>
> ***
> **Q3. Clarification of Algorithm 1.**
>
> Thanks for your suggestion and we will improve the clarity of Algorithm.1. $\hat A$ refers to the desired action indicated by the scoring metric defined in Sec.3.1. Here "$T$ is not satisfied" should be "latency objective $S$ is not satisfied". We are sorry for the notation typo and will fix it in the revision.
>
> For architecture optimization, the model is initialized from pretrained SD-v1.5, and only 'removing' actions ($\hat A^-$ ) are executed, looping until target latency $S$ is satisfied. When the target latency is achieved, the evolving process comes to a vibrate equilibrium. If the latency falls below the objective, the 'adding' action ($\hat A^+$ ) is executed, as in Line 158 in Alg.1.
>
> For both actions ($\hat A^-$ and $\hat A^+$), we add or remove a group of blocks at a time to save evolving and validation costs. For instance, we remove 2 attention blocks in the first Downsampling stage at a time when these blocks are identified as less valuable (slow, contribute less CLIP score).
>
> ***
> **Q4. About the detailed distillation algorithm in Sec. 4.1.**
>
> Thanks for the suggestion. We provide an explicit step distillation algorithm in the author response PDF (Alg. 1) and will also include it in the revision.
>
> ***
> **Q5. How step distillation and cfg loss trade off FID/CLIP score.**
>
> Using our CFG loss for the step distillation, the student model has the probability to mimic two CFG steps of the teacher model with one CFG step. Notice that the CFG has the behavior of using the guidance scale to trade off the FID and CLIP score (as in Fig. 4 Left). Therefore, during the step distillation, the sampled guidance range of the CFG loss and the probability of using CFG loss can influence the FID/CLIP trade-off of the student model, which can be attributed to the FID/CLIP trade-off of the teacher model.
>
> As we can see in Fig. 6(d), using a small range of the CFG guidance scale, such as only 3 as the red line, the student CLIP score can not be improved, since the teacher model only provides supervision of good FID but weak CLIP score under CFG scale as 3. On the other hand, by only using the CFG loss without the vanilla distillation loss, as in the purple line, the student model has a weak FID score, since the teacher model has the behavior of overall good CLIP but weak FID score. Therefore, we use the probability of CFG loss and the range of CFG guidance scale to trade-off the FID/CLIP information from the teacher model, which will then be learned by the student model.
>
> Thanks for the suggestion and we will include the discussion in the revised paper.
>
> ***
> **Q6. About model memory.**
>
> Thank you for raising the discussion. As mentioned in the Limitation section (Line 323), our current scope is the speed optimization of text-to-image models (within two seconds of runtime on mobile devices), and we leave storage size and running memory optimizations as future work. In Tab.6 in Appendix, we can still observe that our efficient model runs with almost 47% less memory on NVIDIA A100 with TensorRT than SD-v1.5. We further obtain the running memory on iPhone 13 Pro Max (iOS 16.5.1), benchmarked by Xcode. Our model requires 3.2G NPU memory (out of a total of 5.5G NPU memory) when using our demo App, while we can not get a consistent estimation of the original SD-v1.5 due to out-of-memory issues on NPU.
>
> ***
> **Q7. Generated images under SD without ResBlocks.**
>
> Thanks for the suggestion. We provide the generated images of SD-v1.5 without ResBlocks before robust training in the author response PDF (Fig.1.b). We will also include them in the revised paper.
>
> ***
> **Q8. About experiments on DiT [a].**
>
> Thanks for suggesting DiT, which is a relevant and great work, and we will discuss it in the revision. In our work, we focus on accelerating large-scale text-to-image models on mobile devices. As for now, DiT only has the model trained on ImageNet. Training DiT on large-scale datasets such as LAION for text-to-image generation requires re-designing the model architecture of DiT to incorporate text condition and many computational resources. We are interested in incorporating architecture improvements from DiT in our future work.
>
> ***
> References:
>
> [a] Peebles, William, and Saining Xie. Scalable diffusion models with transformers. arXiv, 2022.

---

> > ### Comment · Reviewer_vQqa · 2023-08-13
> >
> > I appreciate the authors for the detailed response and the complementary experiments.
> >
> > - The concern regarding the utilization of additional dataset has been well  addressed.
> > - Authors have shown that the work can be extended to other models without heavy hyperparameter tuning.
> > - I now have a clear understanding of Algorithm 1.
> >
> > For the above reasons, I raise my score from 6 to 7.

---

> > > ### Author Response · Authors · 2023-08-16
> > >
> > > Dear Reviewer vQqa,
> > >
> > > Thanks for your prompt response and positive rating! We are glad to know your concerns are addressed, and we will include the additional discussions and analysis in our final manuscript.
> > >
> > > Best,
> > >
> > > Authors

---

### Official Review · Reviewer_5sgC · 2023-07-09

**Soundness:** 3 good
**Presentation:** 4 excellent
**Contribution:** 4 excellent
**Rating:** 8
**Confidence:** 4

**Summary:**

Text-to-image generation (T2I) is getting popular and has a vast application value. This paper explores efficient T2I by reducing computational redundancy. They learn an efficient U-Net via data distillation and then decrease the required diffusion steps. They can achieve T2I on an iPhone 14 Pro in 2 seconds.

**Strengths:**

+ This paper is well-written and easy to follow.
+ The efficiency of T2I models is less explored in previous studies. The proposed efficient U-Net is valuable for future research and can be practical for real-world usage.
+ They provide an attractive demo video in their supplementary.

**Weaknesses:**

+ Why use ViT to encode the input prompt (Fig. 3)? From my best knowledge, ViT is for encoding images, and StableDiffusion borrows the text encoder (similar to BERT) from CLIP.
+ Since they aim at T2I on mobile devices, various devices should be compared/discussed. iPhone 14 Pro is currently one of the most powerful mobile phones. How about those mid-level phones? Or older versions of the iPhone. This study can make this paper more robust.

**Questions:**

Please see the Weakness

---

> ### Author Rebuttal · Authors · 2023-08-09
>
> **We thank the reviewer for the positive feedback and thoughtful comments. We appreciate the reviewer's acknowledgment that the paper proposes a valuable efficient U-Net for research and practical usage, is easy to follow, and provides an attractive demo video.**
>
> ***
> **Q1. About using ViT for encoding the input prompt in Fig.3.**
>
> Here we follow the model notation from CLIP [a] that uses ViT to represent both image encoder and text encoder. In our model, we only use the text encoder from ViT to get the textual embedding from input prompts. Thank you for the suggestion and we will add more details in the revised version to better clarify the implementation.
> ***
> **Q2. Latency on more devices.**
>
> Thank you for the suggestion. We conduct latency analysis on more devices, such as iPhone12 Pro Max and iPhone13 Pro Max, and will add these results in the revision. We use the latency benchmark tool from Xcode to get the reproducible runtime on different devices.
>
> >|      Device      | Text Encoder (ms) | UNet (ms) | VAE Decoder (ms) | Overall (s) |
> >|----------------|:-----------------:|:---------:|:----------------:|:-----------:|
> >|   iPhone14 Pro   |        4.0        |    230    |        116       |     1.96    |
> >| iPhone13 Pro Max |        5.7        |    315    |        148       |     2.67    |
> >| iPhone12 Pro Max |        6.3        |    526    |        187       |     4.40    |
>
> ***
> References:
>
> [a] Radford, Alec, et al. Learning transferable visual models from natural language supervision. ICML, 2021.

---

### Author Rebuttal · Authors · 2023-08-09

Dear Reviewers,

Thank you all for your positive rating and valuable comments. We upload a one-page PDF to include additional figures and algorithms. All other questions are addressed in the following individual responses. We sincerely look forward to having further discussions with you during Reviewer-Author Discussions if there are any other questions.

Thanks,

Authors

---

### Decision · Program_Chairs · 2023-09-21

**Decision:**

Accept (poster)

**Comment:**

All the reviewers recommend the acceptance of the work. Reviewers appreciated the timely work on compressing large generative models and good results. Reviewers raised some clarification questions and also raised concerns regarding the large training time. Authors addressed several of these questions in their responses. The reviewers did raise some valuable concerns and suggestions that should be addressed in the final camera-ready version of the paper, which include adding the relevant rebuttal discussions and revisions in the main paper. The authors are encouraged to make the necessary changes to the best of their ability.